# Determination of Critical Factors for Success in Business Incubators and Startups in East Java

**Habiburrahman [1,\*], Andjar Prasetyo [2], Tri Wedha Raharjo [3], Herrukmi Septa Rinawati [3], Trisnani [4], Bambang Riawan Eko [4], Wahyudiyono [4], Sekar Nur Wulandari [5], Mochammad Fahlevi [6,\*], Mohammed Aljuaid [7] and Petra Heidler [8]**

1 Faculty Economic and Business, Universitas Bandar Lampung, Bandar Lampung 35142, Indonesia
2 Regional Development Planning Agency, Magelang City 56172, Indonesia
3 Regional Development Planning Agency, Surabaya 60174, Indonesia
4 National Research and Innovation Agency, Jakarta 10340, Indonesia
5 Planning, Research and Development Agency, Tanjung Pinang 29113, Indonesia
6 Management Department, BINUS Online Learning, Bina Nusantara University, Jakarta 11480, Indonesia
7 Department of Health Administration, College of Business Administration, King Saud University, Riyadh 11451, Saudi Arabia
8 IMC Fachhochschule Krems, University of Applied Sciences Krems, 3500 Krems, Austria
\* Correspondence: habiburahman@ubl.ac.id (H.); mochammad.fahlevi@binus.ac.id (M.F.)

**Abstract:** The reference to the success factors of incubators and startups in running their business so far has been ambiguous. The purpose of this study is to analyze the critical factors that can affect the success of incubators and startups. The quantitative research method describes the research results. The study locus is in East Java Province, Indonesia with primary data from respondents in five regions: Banyuwangi, Jember, Madiun, Malang, and Surabaya. The number of respondents was 41 from incubators and 59 from startups with identification of domicile, type of business, and business turnover. Data was collected using surveys and interviews with 100 respondents. The analysis in this study uses eleven factors that are assumed to be factors of business success that have been tested with Kaiser Meyer Olkin Measure (KMO), Measure of Sampling Adequacy (MSA), Bartlett's test, and Cronbach Alpha. The critical point used for the KMO value is between 0.5 and 1, the MSA test critical point is 0.5, Bartlett's test is less than the significance level ($\alpha$ = 0.05), and $\alpha$ cronbach $\geq$ 0.60. The analyzed factors are as follows: synergy products; processes; innovation management; communication; culture; experience; information technology; innovation skills; functional skills; and implementation skills. As a result, incubators and startups agree on eleven critical factors to building their business success, but there are differences in the priority scale between incubators and startups on these eleven factors.

**Keywords:** critical success; business; incubators; startups; innovation; management

## 1. Introduction

Business success in business is the final measure for economic actors, generally starting with micro or small businesses that require various determining factors. Scientific attention also appears in several perspectives, such as the competitive advantage approach in the article [1,2] in collaboration with the circular economy and Ref. [3] analysis with information technology capabilities and organizational intelligence. This condition reflects that startups can contribute to technological growth and have a multiplier effect on many businesses. A startup is an organization designed to find the right business model which has become an important issue from a scientific perspective, such as [4] an agile culture combining clan with adhocracy; the ability to nurture their absorptive, innovative, and adaptive capabilities effectively; and a human capital with adequate entrepreneurial skills, emotional attachment to and fitness with the start-up [5]. This indicates that the components of entrepreneurial

intention including knowledge sharing, reputation, social relations, and identity have a positive effect on the performance of digital startups mediated by social media [6]. Meanwhile, adding the word 'digital' is intended for startup companies that integrate digitalization into their products and processes. The word 'startup' has also become very familiar in the entrepreneurial world, but only few know the meaning of the word 'startup' itself. Currently, a startup is defined as a business that is just beginning and it applies technological innovation to run its core business and solve a problem in society. Therefore, it has a 'disruptive' nature in an existing market/industry or even creates a new industry.

The success of the Business Incubator is a business support process that can accelerate the successful development of startup companies so that they can motivate the millennial generation to have new companies ranging from technology, and retail, to media. Although other aspects have appeared in some scientific literacy, sustainable supply chain management [7], strategic improvisation and organizational memory [8], human resources management strategies [9], sustainable supply chain management [10], and smart contracts improve procurement efficiency through cost, time, and quality [11]. Some of the startups are growing and successful, but some of them have had to close their business due to one or two factors that the company did not do well. Previous research related to the determinants of the success of a startup business is as follows: According to [12], acceptable factors to reflect success in a startup business in Surabaya are work ethic, motivation, work discipline, integrity, work involvement, communication, business ethics, and adaptation. An unacceptable factor to reflect on the success of a startup business in Surabaya is the marketing strategy. Ref. [13] stated that the determinants of business success for startup businesses at Tyfons, Tlab, and Icube startups, were good human resources/teams, right timing, as well as sufficient ideas and funding. An agile culture combining clan with adhocracy; the ability to nurture their absorptive, innovative, and adaptive capabilities effectively; and human capital with adequate entrepreneurial skills, emotional attachment to, and fitness with the startup [4]. It harmonizes with their agile culture, effectively enabling innovation programs enabled startups to remain focused on their innovation initiatives and not worry about scalability since the solutions collaborations between employees internally and with external actors enable rapidness to the market [5]. The research gap in research tries to solve the issue of, in Indonesia today, there are thousands of startups that have spread throughout the region. Competition always haunts startup businessmen. This will be difficult for new startups, if they do not have strong business knowledge. Many new startups fail in their development, due to poor business planning, and are unable to compete with other market competitors and lack of investor funds. Therefore, new startups need an incubator program in building a business to be more focused and ready to be launched into the community, from developing a business to getting investors. Successful companies are constantly creating and distributing new knowledge and rapidly applying it to new technologies and products [6]. From these several literacies, this study explores the factors needed in incubators and startups, according to certain conditions, and analyzes them into critical success factors.

## 2. Literature Review

The theoretical foundation is a concept in the form of a statement that is neatly arranged and systematically has measurable variables in research because the theoretical basis will be a strong foundation for research to be carried out [14]. In this section, the theoretical basis used in this research will be explained. The theoretical basis used is based on the thoughts of previous experts and researchers to formulate research models based on existing theories to be tested empirically by the following statistical rules.

In compiling a research with a quantitative approach method, it is necessary to order the theories that will be used systematically starting from the Grand Theory, Middle Range Theory, and Applied Theory. Grand theories in general are macro theories that underlie the various theories below. The term grand theory is used because the theory is the basis for the birth of other theories at various levels. Grand Theory is also called macro because

these theories are at the macro level, talking about structure rather than micro phenomena. Middle theory is where the theory is at the middle level which focuses on macro and micro studies; whereas Applied Theory is a theory that is at the micro level and is ready to be applied in conceptualization [15].

### 2.1. Resource Based Theory (RBT)

Resource Based Theory (RBT) is a managerial framework used to determine the strategic resources that a business can leverage to achieve a sustainable competitive advantage [16]. Barney [17,18] is widely cited as an important figure in the emergence of the resource-based view. RBT proposes that a business is heterogeneous because every company has heterogeneous resources, so every business such as a startup can have a different strategy because they have different resource capacities [17]. RBT focuses managerial attention on the company's internal resources in an effort to identify the company's resource assets, capabilities and competencies with the potential to provide a superior competitive advantage [19]. Twenty years later, resource-based theory is widely recognized as one of the most prominent and powerful theories for describing, explaining, and predicting organizational relationships [20]. RBT has undergone an evolution that reflects the first three stages of life cycle theory, namely, recognition, growth, and maturity [21]. Although some previous research has identified organizational resources as important, RBT was not widely discussed until the 1980s. The decade was dominated by externally focused frameworks, such as Michael Porter's Five Forces Model in 1980, but the emergence of RBT has gradually begun to shift the attention to management discussions [22].

Based on this insight, the resource-based view (RBT) assumes that successful businesses and startups are driven by the capabilities and competencies of the firm's resources, hence the firm's resources are more important for determining strategic actions than its external environment [23]. This approach takes an inside-out view of strategy. In this situation, the startup is better known to planners and internal data is usually more readily available. RBT is an inside-out perspective on organizations that seeks to identify the characteristics of businesses with superior performance. The main idea is to build on internal strengths rather than relying on external strengths, because by the time a business prepares to absorb those external strengths, those strengths may already be left behind. RBT understands each business as a unique pool of resources that fall into three categories: tangible assets, intangible assets, and capabilities [22]. Tangible assets (e.g., financial and physical) and intangible assets are resources owned by a business (e.g., intellectual property, organizational assets, reputation), and capabilities are what a company can do (e.g., its knowledge). Resources and abilities are thus different constructs.

Resources in the form of goods can be traded and are not specific to a particular company; whereas, capabilities are specific to a particular company (because they reside in people) and are used to involve resources within the company. In some studies, a strict distinction between resources and capabilities is not necessary [20,24]. In this study, it is also not necessary to distinguish "strategic" resources from others, because most of the resources are actually easy to imitate or trade. It is understood that some capabilities are more complex and are created by combining resources in the form of complex goods and capabilities (knowledge). In RBT, these resources and capabilities are the main determinants of competitive advantage, and strategic planning must start from these resources.

### 2.2. Dynamic Capability

According to [25], Dynamic Capability is the ability to integrate, build, and reconfigure internal and external competencies to cope with a rapidly changing environment. Dynamic Capability extends resource-based theory by emphasizing the role of processes/routines in achieving a competitive advantage. The dynamic capabilities view has emerged as a central approach to answering the question of how a business copes with technological change. Capturing the essence of dynamic capabilities and understanding what they are and how they actually support technological innovation and change, however, has

until recently been a formidable challenge [26]. This study discusses the role of startups that involve a lot of technology in improving their business through dynamic capabilities theory. Technological changes continuously create new challenges and opportunities, new products, services, processes, and business development. However, these opportunities need to be captured and converted into value through effective and dynamic management. This process requires a new way of understanding technology management that captures its dynamic nature as well as its managerial aspects. The business factors success factor framework is based on dynamic capability theory, which emphasizes the development and exploitation of continuously changing technological capabilities [27].

Dynamic capabilities theory is not always concerned with fixed assets, but rather aims to explain how startups allocate resources for innovation over time, how startups generate and deploy existing resources, and where they acquire new resources. This is particularly relevant for developing approaches to success factors that can explain how the combination of resources and processes can be developed, deployed, and protected for each startup activity. A framework is proposed to position startup activities within the broader business context, supported by case studies to illustrate the value of the business framework itself. Three dynamic abilities are needed to meet new challenges. Startups and their employees need the ability to learn quickly and build strategic assets. New strategic assets such as capabilities, technology, and customer feedback must be integrated within the company. Existing strategic assets must be changed or reconfigured. Dynamic capability concept basically saying that what matters to a business is startup agility: the capacity to (1) sense and shape opportunities and threats, (2) seize opportunities, and (3) maintain competitiveness through upgrading, merging, protecting, and, where appropriate, configuring the reclaim of intangible and tangible assets of a startup business. Collaborations and partnerships can be a source of learning for new startups, helping companies recognize dysfunctional routines and prevent strategic blind spots. Similar to learning, building strategic assets is another dynamic capability. For example, alliances and acquisition routines can allow startups to bring new strategic assets into the company from external sources [28]. Effective and efficient internal coordination or strategic asset integration can also determine startup performance. According to [29,30], quality performance is driven by startup-specific routines for gathering and processing information, linking customer experience to engineering design choices, and coordinating component manufacturers and suppliers. Competitive advantage also requires the integration of external activities and technologies. Ref. [31] shows that internal and external human resources and technology resources are closely related to the commercialization of technology.

A fast-changing market requires the ability to reconfigure the startup asset structure and achieve the necessary internal and external transformations [32]. Change is expensive and so startups have to develop processes to find high-yield change at a low cost. The ability to change depends on the ability to scan the environment, evaluate the market, and quickly complete reconfiguration and transformation in the face of competition can be supported by decentralization, regional autonomy, and strategic alliances [33].

### 2.3. VRIO Framework

In the resource-based literature, the VRIO framework of value-rarity-imitability organization [34] has become the most widely recommended method for valuing a particular firm's resources. This technique stems from the development of RBT theory and was not originally a tool for practical application. Subsequent developments as a means of understanding enterprise resources have helped VRIO to spread widely. In comparison, the company-specific focus, as articulated in the company's RBT, focuses on the company's privileged resources [35]. In this approach, startups are described as a collection of tangible and intangible resources, whereas strategy selection and implementation is based on careful evaluation and leverage [36]. A startup's strategic goal is to develop and deploy a combination of valuable and rare resources that cannot be imitated, substituted, or purchased directly by competitors. If this goal is achieved, further innovation advantages can be built

and maintained. Thus, in trying to explain the variation of innovation, RBT argues that researchers should investigate directly the resource base of startups and not the structural characteristics of the industry [37].

Startups can identify which of these resources and capabilities are capable of creating a sustainable competitive advantage, and [38] set four criteria for solving this question. For a resource or capability to be strategically useful, it must be valuable, rare, inimitable, and non-substitutable. In addition, [39] argues that to determine whether a capability is a core competency—the basis for a startup's competitive advantage—the capability must provide "potential access to multiple markets" and must significantly increase the benefits of the final product or service as perceived by the customer. These two criteria together determine whether a resource or capability is "valuable". Another test is that the resource or capability must be relatively scarce compared to its demand, difficult for competitors to imitate, and (as a special case of non-replicability) cannot be replaced by other resources or capabilities that competitors might develop. Startups must also be able to capture this advantage in order to succeed.

*2.4. Technology Capability*

Technology Capability has been described as the startup's ability to design and develop novel processes, products, and enhance knowledge and skills about the physical environment in a unique way and transform knowledge into instructions and designs for efficient creation of the desired performance [40]. Startup technological capabilities include not only technical mastery capabilities, but also the capacity to expand and deploy the startup's core capabilities and effectively incorporate various technology segments and mobilize technology resources across the enterprise [41]. According to [42], technological capabilities are the technical, managerial, and institutional information and skills that enable productive enterprises to utilize equipment and technology efficiently. In the current phenomenon, technological capabilities exist in both general and company-specific sectors, technological capabilities become a form of institutional knowledge consisting of combined skills that are accumulated by its members over time and technology development is a process of building these capabilities. Technological capabilities consist of practical, theoretical knowledge, procedures, experience, methods, equipment, and physical devices. Technological capabilities represent superior startup technical resources that are closely related to design technology, product technology, information technology, and external knowledge integration. This technology capability component is responsible for the significant positive variation in startup performance [43].

Technological capabilities enable startups to identify, acquire, and apply new external knowledge to develop operational competencies that lead to achieving superior performance [44]. Through effective technological capabilities, startups can create and deliver new products and services in a better and efficient manner that best satisfies consumer needs, thereby increasing the overall success of the company's new product development and performance [45]. At this point, startups need to pool the resources and competencies that enable them to have more developed technological capabilities than their competitors. In this case, technological capability relates to the absorption and transformation of technology as a way to achieve a higher level of technical-economic efficiency [46]. There is a large body of literature linking technological capabilities to business knowledge and innovation [47]. Ref. [48] explained that companies develop their technology capabilities gradually and in doing so, they are limited to continuing to do what they already know, which means there is a cognitive limit to what startups are capable of. In short, the concept of technological capability includes the generation of new knowledge and learning [49].

Startups innovate because they hope to gain economic benefits for the company. In some cases, these benefits do not come from new product launches. The process can actually come in stages, from adjustments in the production process, in organizational structures, or even from marketing actions. They all make it possible to create higher margins for the company. An understanding of the company's performance develops, along with

the relationship of its technological capabilities, with the development of technological advances. Innovation can be associated with reduced costs and higher profits, as technology becomes more and more complex, startups need to have a way of keeping and running their business to keep up with developments [50].

## 3. Methods

A descriptive method with a quantitative approach is used in this study, with a time starting from February to September 2021. The scope of the study is in the work area of the Center for Human Resources Development and Research in Communication and Information Technology, Surabaya, East Java, Indonesia. This data collection is based on a government program in recording business data in each region in the province of East Java so that this data is taken collectively and simultaneously [51]. Primary data were obtained from 100 people consisting of managers of 41 incubators and managers of 59 digital startups spread across the cities of Banyuwangi, Jember, Madiun, Malang, and Surabaya as respondents in this study. Characteristics of the selection of respondents are based on the demographic criteria of domicile location, type, and business turnover. The data collection instrument used interviews, observations, and questionnaire guidelines. The questionnaire was answered quantitatively with a Likert scale whose value was limited from one to five [52]. A value of one means very uncertain; a value of two means undefined; the value of three means quite decisive; the value of four means determining; a value of five means very decisive [14]. Data collection methods include (1) participation, direct involvement with incubator managers and digital startup managers; (2) Focus Group Discussion, to filter problems according to the focus of the study. The analysis is based on the success factors, in this study, among others: (1) synergy; (2) product; (3) process; (4) innovation management; (5) communication; (6) culture; (7) experience; (8) information technology; (9) innovation skills; (10) functional skills; and (11) implementation skills. To measure the adequacy of the sample and its significance value, we used KMO Measure, MSA, and Bartlett's test. The critical point used for the KMO value is between 0.5 and 1, while for Bartlett's test it is less than the significance level ($\alpha = 0.05$) [51]. The test results at this point are described as follows:

These results (Table 1) indicate that factor analysis is appropriate to use to simplify the eleven startup success factors and formulate strategic recommendations to accelerate startup success. To determine the accuracy and reliability of the questionnaire in measuring respondents' perceptions, the study instrument was tested [53]. Tests using the provisions of items and indicators of a research instrument that can be declared valid are having a positive correlation value greater than 0.30 ($r > 0.30$). The provision of a research instrument is declared reliable if it has a Cronbach Alpha value exceeding 0.60 ($\alpha$ cronbach $\geq 0.60$). The description of the indicators of each success factor along with the results of instrument testing is presented in Figure 1.

**Table 1.** Results of Measurement of Sufficiency of Samples and Significance.

| Criteria | Result | | Critical Point | Explanation |
|---|---|---|---|---|
| | **Incubator** | **Startup** | | |
| KMO | 0.759 | 0.891 | 0.50–1.00 | Fulfill |
| Bartlett's Test | 0.000 | 0.000 | <0.05 | Fulfill |

Source: Prepared by the writing team, 2021.

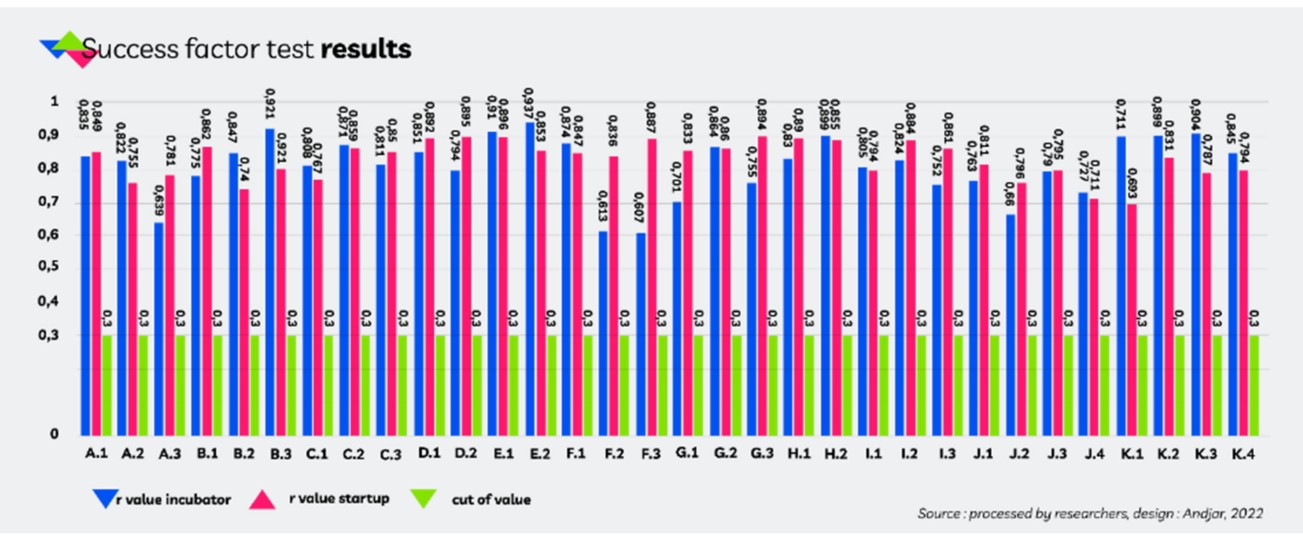

**Figure 1.** Success factor test results.

Referring to the test results of the instrument as shown in the table above, the overall value of r and Cronbach's Alpha meets the critical value as the limit for determining an instrument to be valid and reliable. This shows that the instrument is declared valid and reliable; in other words, respondents understand part by part the statement of startup success factors, and there is a match between respondents' perceptions with the purpose of this study.

## 4. Results

As defined by the National Business Incubator Association (NBIA), a Business Incubator is a business support process that can accelerate the successful development of startups and startups by providing entrepreneurs with the necessary resources and services. These services are typically developed or managed by incubator management and are offered both within the Business Incubator itself and through a network owned by the Business Incubator. Business incubators usually provide programs for early stage entrepreneurs or startups, which are designed to foster and accelerate the success of business development through a series of capital programs followed by partnership support or through coaching other business elements to turn the business into a profitable company, and have management over proper organization and finance, as well as being a sustainable company, so that it has a positive impact on society.

### 4.1. Respondent Character

A total of 100 people consisting of 41 incubators and 59 startups in the cities of Banyuwangi regency, Jember regency, Madiun regency, Malang regency, and Surabaya city as respondents in this research study have characteristics based on the demographic criteria of domicile location, type and business turnover. As many as 24% of incubators are domiciled in Banyuwangi regency and Malang regency, 22% in Madiun regency, 15% in Jember regency, and 12% in Surabaya city, and the rest did not answer. The majority of incubators domiciled in Jember are representatives of the UPT Kewirausahaan dan Inkubator Bisnis Teknologi (KIBT) Banyuwangi State Polytechnic and Poliwangi. The Jember incubator came from the Politeknik Negeri Jember and Primadhani. Madiun came from the Politeknik Negeri Madiun c, Malang came from the Politeknik Negeri Malang, and Surabaya came from the Inkubator Bisnis Politeknik Elektronika Negeri Surabaya. Startups as research subjects other than the incubator in this research study are also domiciled in these five cities with the majority domiciled in Banyuwangi (27%), Malang, and Jember (20% each), and the rest are domiciled in Madiun (14%) and Surabaya (12%). The types of businesses that are currently being carried out are also diverse, including the fields of

food, beverage, handicrafts, services, and information and technology fields, one of which is the development of community security applications in Madiun City, property services, aquaculture and agriculture, convection, construction, even in the field of education. The variety in types of business also, of course, accompanies the variety in turnover in for each type of business. Detected based on the domicile area, Madiun and Surabaya excel with an estimated total business turnover from less than IDR 300 million (IDR ≤ 300 million) to IDR 4.5 billion. In more detail, Madiun includes services in the creative industry, application creation services, and property services. Surabaya with the types of business services, manufacturing, IT, aquaculture, and agriculture, where Surabaya is the second biggest city on the island of Java, which has the highest turnover range compared to startups domiciled in the other three regions (Banyuwangi, Jember, Malang). Jember with its startups, covering the types of food, beverage, manufacturing, and handicraft businesses, has a turnover ranging from IDR 300 million to IDR 500 million.

Meanwhile, Banyuwangi and Malang each have a type of business with a turnover of less than IDR 300 million with the types of business in food, beverage, convection, creative industry, software house, fertilizer, animal feed, and construction (Figure 2).

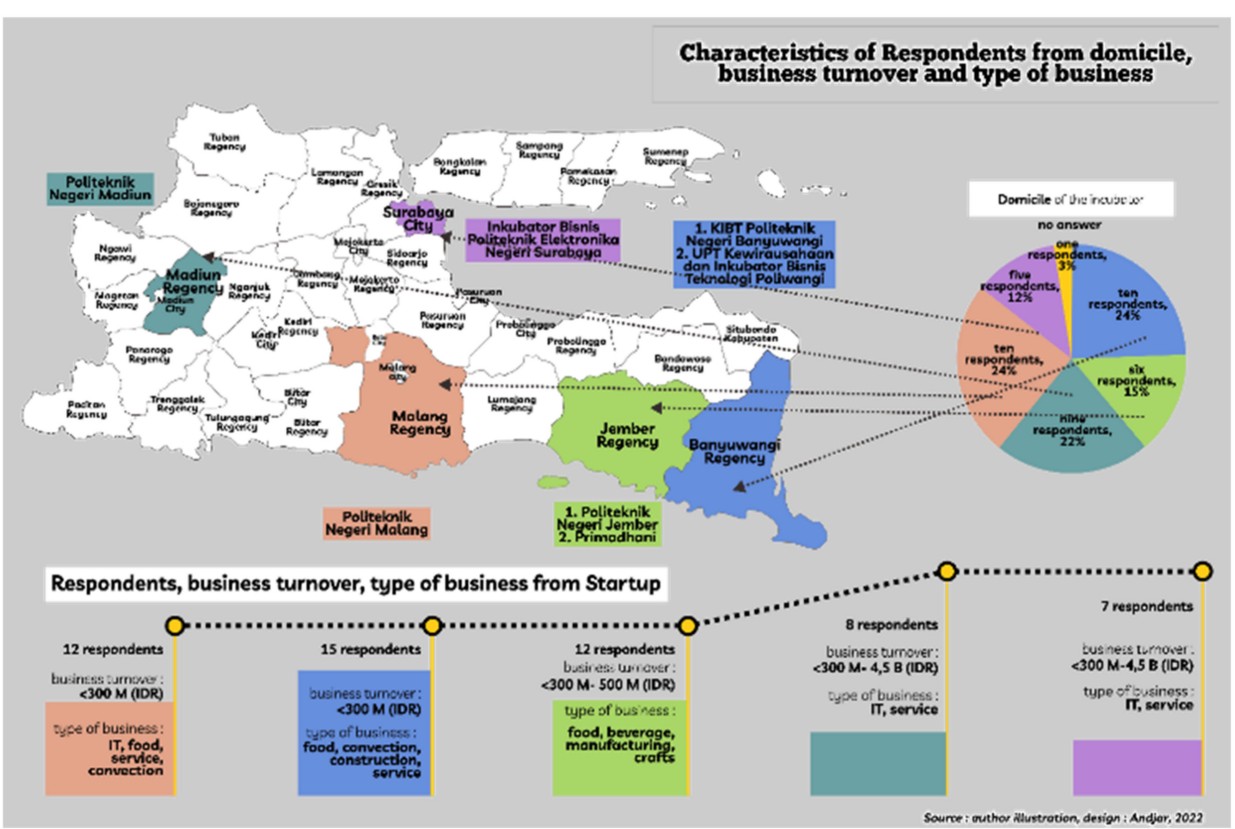

**Figure 2.** Characteristics of Respondents from domicile, business turnover, and type of business.

### 4.2. Determinants of Incubator and Startup Success based on Respondent Priority Scale

Incubator and Startup respondent groups who are domiciled in five cities in this research study both agree that the eleven determinants of startup success need to be implemented in entrepreneurship. However, in practice, both incubators and startups have their priorities.

Regarding the priority scale according to the incubator, product is the main priority scale because it is a vocational-based institution. As product is the main thing, innovative and valuable products have a selling value in the market, and as the main factor in startup development, success is measured by the level of sales; the second priority is innovation skills because innovation makes a product survive in the market, is used in the context of

product development, its application is carried out after careful planning, and innovation is what makes a product unique. The third priority is process and communication because the process supports the success, development, and marketing of products and determines production. Meanwhile, communication is the main means of conveying ideas and actions, and with clearer direction and communication, we can find solutions to problems that have the potential to hinder startups, as well as being a medium for creating teamwork. The three priority scales are compiled based on facts that have been known by the incubator as a companion for startup actors in entrepreneurship (see Figure 3). Thus, product, innovation skills, process, and communication are four of the eleven determinants of startup success which, according to the incubator's observations, have proven to be the determinants of success and have been implemented by startup actors.

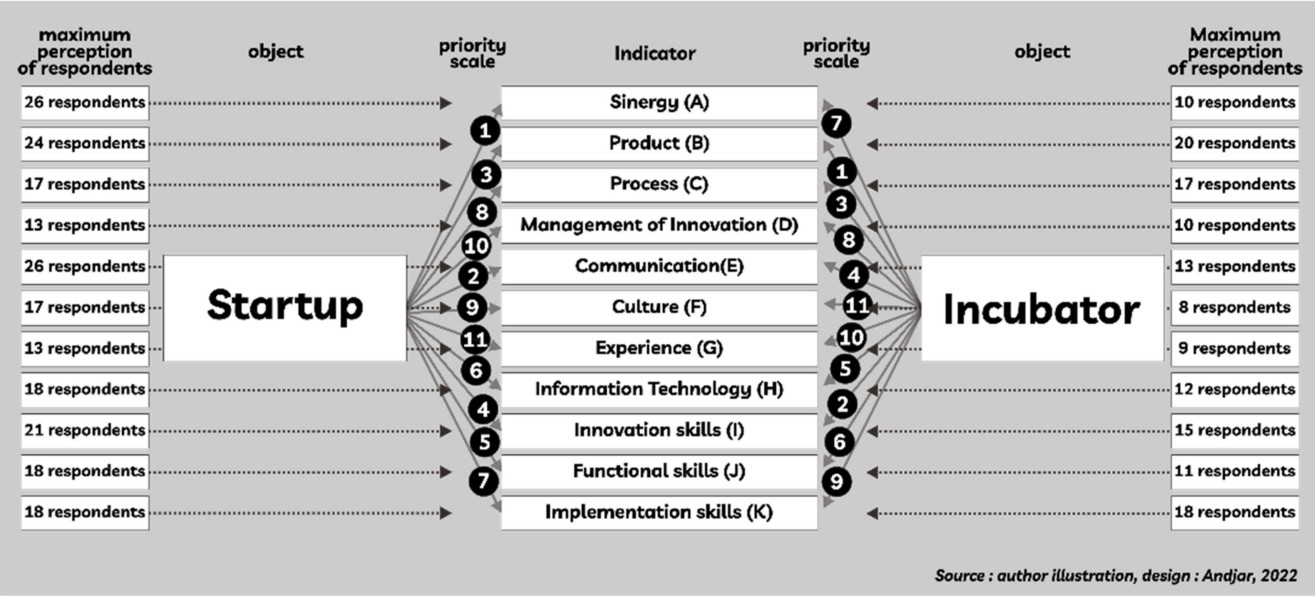

**Figure 3.** Priority Scale for startup and incubator versions.

Regarding the priority scale according to startup, synergy and communication are the main priority scales because they can improve the performance of each line, provide a benchmark for business stability and sustainability, and be the basis for implementing the vision and mission to reach targets, accelerate startup development progress and build individual trust in the team. Terms of communication are an important component because it is a means of initial interaction in forming a solid team and supporting the smooth production process. The second priority is the product because it is an important component, as well as business support and startup icon, while the third priority is innovation skills that make startups superior and competitive. Thus, synergy, communication, product, and innovation skills are four of the eleven critical success factors that have been implemented by startup actors (see Figure 4).

### 4.3. Factor Analysis

The factor analysis in this research study is used to identify and evaluate the influencing factors and strategies for accelerating the success of digital startups. Furthermore, it is needed to find out whether the eleven business success factors used in this study are adequate for further analysis. The test uses the MSA with the limit of the anti-image correlation matrix value with a critical point of 0.5 which must be passed by the eleven factors.

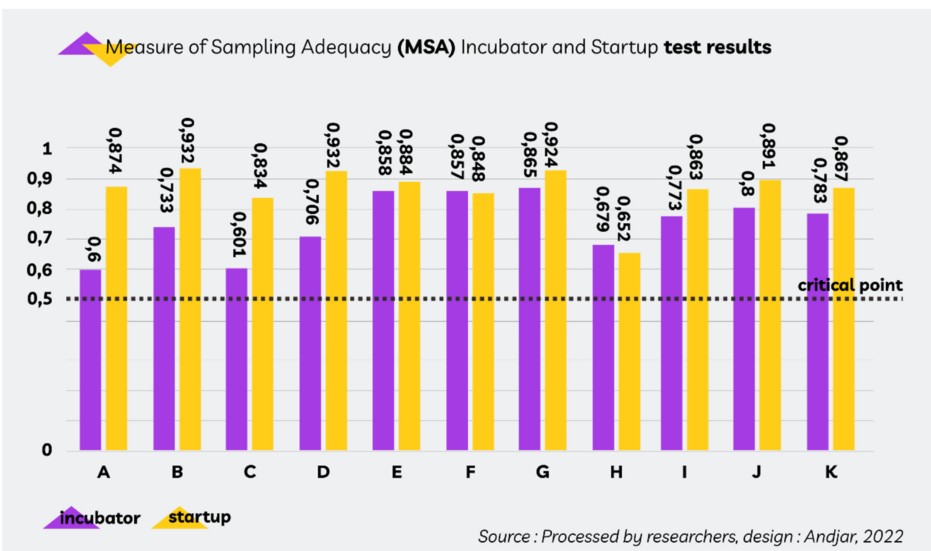

**Figure 4.** MSA Incubator and Startup test results.

These results indicate (Figure 4) that the anti-image correlation matrix value of the eleven factors for both incubator and startup respondents has exceeded the critical point of 0.500. In other words, eleven factors measured from the incubator's and startup's point of view were deemed adequate for further analysis.

### 4.4. Simplification of Startup Success Factors from the Incubator and Startup Perspective

There are three simplification groups of the eleven success factors of digital startups based on the incubator's point of view, whereas from the startup's point of view, only one factor is formed. Further identified, the three factors formed based on the results of the 41 incubator's perception measurement, it can be stated that factor 1 to factor 3 determines the urgency or priority factor chosen as a strategy in accelerating the success of digital startups. Factor 1 is declared as the top priority, factor 2 is the second priority, and factor 3 is the third priority. Meanwhile, from the startup point of view, the eleven success factors for startups are inseparable units in the implementation of entrepreneurship. These eleven factors are considered very important and, in the process, must be implemented together or simultaneously.

It can be seen that the incubator divides the eleven determinants of startup success into three priorities (Figure 5), while startups do not divide the eleven factors into several priority levels. This shows that from the point of view of the incubator as a guide in the implementation of entrepreneurship, it provides three levels of urgency that can be used as a reference by startups in entrepreneurship and proceed step by step toward the point of success.

Meanwhile, the startup does not break down the eleven factors into several levels of urgency like the incubator, meaning that the startup views the eleven factors as an integral part that cannot be separated from its implementation from the initial process to finalization in one rotation of startup business activities. Although in the process, there are several of the eleven factors that cannot be carried out simultaneously, and in concept, they are interrelated and influence each other.

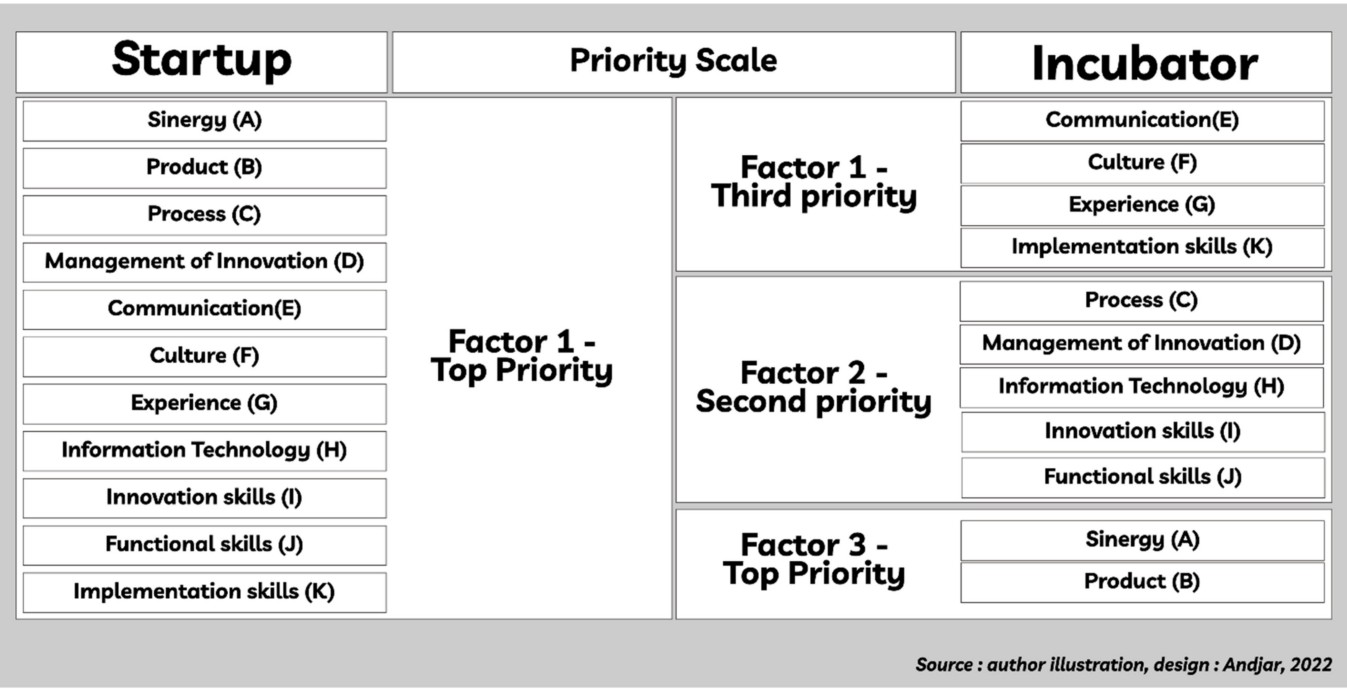

**Figure 5.** Simplification of Factors in the priority scale.

## 5. Discussion

So far, the determinants of startup success are determined based on the priority of the interests of the main components and supporting processes, development, and marketing. According to the Incubator Product point of view, innovation skills, process, and communication are four of the eleven determinants of startup success according to observations. The incubator has proven to be a determinant of success and has been implemented by startup actors. Meanwhile, according to digital startup business actors, synergy, communication, product, and innovation skills are four of the eleven critical success factors that have been implemented so far.

The difference in priority scale between the incubator and startup perspectives is a natural thing because startup business actors know the details from the planning, process, to the production marketing stages [29]. This makes business actors have preferences ranging from the main factors to alternative factors that are used as a reference in the entrepreneurship process, in addition to getting guidance from the incubators. Examining in more detail with factor analysis, the eleven factors that support the success of digital startups are simplified into three parts, which are then identified as a scale of urgency in further implementation according to the incubator's point of view. Meanwhile, according to startup, it is not simplified and instead includes eleven factors into one main priority [30]. These conditions are described in detail to form a strategy to accelerate the success of startups that is compiled from the point of view of the incubator as an evaluator and the startup as a business actor (see Table 2).

**Table 2.** Strategies to Accelerate Startup Success.

| Incubators | Startups |
|---|---|
| **Main Priority** | **Main Priority** |
| 1. Synergy—coordination in improving the quality of knowledge and skills of startup members.<br>2. Products—products that are easy to access and use. | 1. Synergy—coordination in improving the quality of knowledge and skills of startup members.<br>2. Products—products that are easy to access and use.<br>3. Process—(a) management of clear boundaries between products and services in startup development with a strategy of integrating product components as a whole and digital services; (b) submission of information related to the digital channel used (for example, software platforms, operating systems, web services) |
| **Second priority** | 4. Management of innovation—implementation of consistent innovation management with problem-solving strategies and good planning |
| 1. Process—management of clear boundaries between products and services in startup development with a strategy of integrating product components as a whole and digital services.<br>2. Management of innovation—implementation of consistent innovation management with problem-solving strategies and good planning<br>3. Information Technology—utilization of information technology to the maximum as a supporter of the main and supporting products.<br>4. Innovation skills—have the basic skills, knowledge, abilities, and personal qualities needed for success in the workplace (thinking skills, personal qualities, time management, and interpersonal skills).<br>5. Functional skills—understanding of internet technology. | 5. Communications—communication plays an important role in reflecting the value of the product and increasing the number of users<br>6. Culture—(a) structured work rhythm with balanced flexibility; (b) coordination mechanism in achieving improvisation and innovation<br>7. Experience—(a) records of previous experiences as material for evaluating further innovations; (b) specific experience in technology skill development, production, marketing |
| **Third priority** | 8. Information Technology—utilization of information technology to the maximum as a supporter of the main and supporting products |
| 1. Communications—communication plays an important role in reflecting the value of the product and increasing the number of users.<br>2. Culture—coordination mechanism in achieving improvisation and innovation.<br>3. Experience—specific experience in technology skill development, production, and marketing.<br>4. Implementation skills—startup and product standards knowledge and compliance. | 9. Innovation skills—have the basic skills, knowledge, abilities, and personal qualities needed for success in the workplace (thinking skills, personal qualities, time management, and interpersonal skills)<br>10. Functional skills—understanding of internet technology<br>11. Implementation skills—startup and product standards knowledge and compliance |

Source: Prepared by the writing team, 2021.

From the previous description, the strategies that need to be implemented to accelerate the success of digital startup businesses are divided into stages according to development priorities. The top priority, the most important and main strategy to be taken immediately in accelerating the success of a startup, is related to two things: synergy and product. Synergy—building synergies is related to how each member of the company can help each other and share to improve the quality of knowledge and skills, which is done by sharing knowledge through cross-technology, cross-hierarchical and cross-functional exchanges [54]. Regarding synergies, the strategies that need to be implemented are: providing an understanding of startups, that to build a business, good cooperation is needed with a common vision and mission of the company so that knowledge and skills are needed in developing a business through the creation of products and services, both those that are general or by the field and position handled. Product—product development needs to be carried out by startups to produce products that are easily accessible, easy to use, and are consistent with the aesthetics of a digital product service, so that product users will feel the ease and comfort of using the product. Regarding products, the strategies that need to be implemented are: increasing understanding of product knowledge, especially related to design, features, brand names, product variations, quality, services, packaging, and returns, as well as providing ease of access and use, thus providing convenience and convenience for users [55].

The second priority, in accelerating startup success is related to five variables, including process, management of innovation, information technology, innovation skills, and functional skills, so the strategies implemented are also related to these variables. Process—how startups run the product development process from concept to customer to provide solutions needed by customers, which involves: (a) bundling, combining components of digital product and service portfolios to provide clear boundaries and relationships between products and services; (b) devices, information about hardware and software used in the development phase; and (c) channels, related to the use of digital information channels, which include: software platforms, operating systems, and web services. Regarding the process, the strategies that need to be implemented are: building a good understanding for startups in product development from concept to product to customer that provides solutions according to customer needs, which include: (a) merging digital product and service components; (b) use of software and hardware in development; and (c) the use of software, operating systems, and web services. Management of innovation— how startups can manage innovation so that they can run the product development process quickly while reducing the uncertainty that will occur which includes: management style, consistent stages of innovation management, problem-solving and good planning. Regarding the management of innovation, the strategies that need to be implemented are: building knowledge and skills in managing innovation development, through a productive management style, good planning, consistent innovation, and emphasis on problem-solving. Information Technology—related to information technology is how startups can use information technology to support their business success, either as the main product or as a support. Regarding information technology, the strategies that need to be implemented are: increasing the understanding and skills of startups in utilizing information technology to support business success. Innovation skills—innovation skills involve skills that should be possessed by a good startup: basic skills, academic skills, technical skills, generic skills, soft skills, managerial skills, and entrepreneurial skills that contribute to innovation. Regarding innovation skills, the strategies that need to be implemented are: developing skills, that can be basic, academic, technical, or generic, soft skills, management, and entrepreneurship that supports innovation. Functional skills are related to how startups can use the information and digital technology, in the form of (a) the ability to use software and hardware; (b) understanding of internet technology; (c) understanding of hardware/system architecture; and (d) ability to troubleshoot software and hardware problems. In terms of functional skills, the strategies that need to be implemented are: developing startup skills in using information and digital technology which includes: (a) software and hardware; (b) internet technology; (c) system architecture; (d) solving software and hardware problems [56].

The third priority, or final priority in accelerating startup success, is related to four variables, namely: communication, culture, experience, and implementation skills. Communication is about how startups inform the value of products and industry service standards so that they can be quickly accepted by the industry, market, and potential consumers, where this communication can use online or offline channels [57]. Concerning communication, the strategies that need to be implemented are increasing understanding and skills in understanding the value of products and service standards accepted by the industry, market, and consumers to be delivered online and offline. Culture is related to how startups create a comfortable and conducive working atmosphere to support company members to interact and communicate to make startups have a solid team [58], which includes: (1) structured, flexible, and balanced work rhythms; (2) the availability of a special time for improvisation and innovation by each employee; and (3) the existence of a coordination mechanism in the process of improvisation and innovation. Concerning culture, the strategies that need to be implemented are: increasing the understanding of startups in building a comfortable and conducive work atmosphere through interaction and communication between members of the startup team related to (a) work rhythm and (b) improvisation and innovation. Experience in previous projects allows the company to develop the ability to improve subsequent innovations so that they can develop skills in technology, production,

and marketing, which enable the use of learning by acquiring advantages and maximizing the experience in managing the organization or producing better products. Regarding the experience, the strategies that need to be implemented are increasing understanding of the use of experience to support the learning by completing the process of managing the business. Implementation skills—this is the startup's ability to anticipate and prepare policies related to product implementation used by users, including (a) knowledge of company practices and organizational protocols; (b) understanding and effective use of industrial terminology in product implementation; (c) knowledge of and compliance with the requirements of industry norms; and (d) knowledge of and compliance with company and product standards. Regarding implementation skills, the strategies that need to be implemented are: improving startup capabilities related to (a) company practices and organizational protocols; (b) use of industrial terminology in product implementation; (c) knowledge of and compliance with the requirements of industry norms, as well as company and product standards.

One of the Indonesian government's programs for business incubators and startups is to provide training to nurture the talents of digital startup founders. The full activity is completed online. Preparing for mentoring and becoming a school for founders (startup founders) can be accessed anywhere in tier 1, tier 2, or tier 3 cities. In the startup incubator, there are many activities that are usually carried out, starting from introducing startups in front of investors, to introducing products and even introducing the startup business model itself. Then, the startup gets a direction and guidance to be able to perfect the concept, product, marketing and various things that can accelerate the growth of the business. Usually, the startup incubator program will take approximately six months. This is because not only startups that already have businesses that will be incubated, but startups whose products are still in the form of concepts or ideas will also be incubated. Through this program, the Ministry believes there will be more accelerated start-ups that can contribute to increasing the number of unicorns and even decacorns in Indonesia, which accelerates the national digital transformation process. With the programs that have been presented, it is hoped that more people in Indonesia, especially the younger generation, are interested and actively participate in creating businesses by utilizing advanced technology and digitalization.

## 6. Conclusions

According to Inkubar's point of view, the factors that influence the success of digital startups are eleven success factors with three different priorities. The priority levels are as follows: the top priority consists of two factors, namely: synergy and product; the second priority consists of five factors, namely: process, management of innovation, information technology, innovation skills, and functional skills; the third priority consists of four factors, namely: communication, culture, experience, and implementation skills. Meanwhile, in the view of the incubator, all factors have the same priority in supporting the success of digital startups, which include the following factors: synergy, product, process, management of innovation, communication, culture, experience, information technology, innovation skills, functional skills, and implementation skills.

Strategies that need to be implemented to accelerate the success of digital startup businesses are divided into stages according to development priorities that are the incubator's concern. Top priority, the most important and main strategy to be taken immediately in accelerating the success of a startup, is related to two things: synergy and product. Synergy factor strategy to build a business requires good cooperation with the same vision and mission of the company so that knowledge and skills are needed in developing business through the creation of products and services, both generals in nature and by the fields and positions handled. Product factor strategy increases understanding of product knowledge, especially related to design, features, brand name, product variety, quality, service, packaging, and returns, as well as providing ease of access and use, thus providing convenience and comfort for users.

Second priority, the second priority in accelerating startup success is related to five variables, including process, management of innovation, information technology, innovation skills, and functional skills. Building a good understanding for startups in product development from concept to product to customer that provides solutions according to customer needs, which include: (a) combining components of digital products and services; (b) use of software and hardware in development; and (c) the use of software, operating systems, and web services. Building knowledge and skills in managing innovation development, through productive management style, good planning, consistent innovation, and emphasis on problem-solving. Improving the understanding and skills of startups in utilizing information technology to support business success. Develop skills: basic, academic, technical, generic, soft skills, management, and entrepreneurship that support innovation. Develop startup skills in using information and digital technology which includes: (a) software and hardware; (b) internet technology; (c) system architecture; (d) solving software and hardware problems.

The third or final priority in accelerating startup success is related to four variables, namely: communication, culture, experience, and implementation skills. Improving understanding and skills in understanding the value of products and service standards that are accepted by the industry, markets, and consumers to deliver online and offline. Improving the understanding of startups in building a comfortable and conducive working atmosphere through interaction and communication between members of the startup team related to (a) work rhythm, and (b) improvisation and innovation. Increasing understanding of the use of experience to support the learning by doing the process of managing a business. Improve startup capabilities related to (a) enterprise practices and organizational protocols; (b) use of industrial terminology in product implementation; (c) knowledge of and compliance with the requirements of industry norms, as well as company and product standards.

The recommendation from this research is the need for educational and training activities for startups carried out by the Business Incubator or Techno Park. The activities are carried out to take into account the priority scale so that the success of startups can be achieved gradually, according to the level of importance and the availability of the education and training implementation budget.

**Author Contributions:** Conceptualization, H. and A.P.; methodology, M.F.; software, T.W.R. and H.S.R.; validation, T., B.R.E. and W.; formal analysis, S.N.W.; investigation, A.P.; resources, M.F.; data curation, A.P.; writing—original draft preparation, H. and A.P.; writing—review and editing, M.A.; visualization, P.H.; supervision, M.F.; project administration, M.F. and A.P.; funding acquisition, M.A. All authors have read and agreed to the published version of the manuscript.

**Funding:** The authors would like to extend their appreciation to King Saud University for funding this work through the Researcher Supporting Project (RSP2022R481), King Saud University, Riyadh, Saudi Arabia.

**Institutional Review Board Statement:** Not applicable.

**Informed Consent Statement:** Informed consent was obtained from all subjects involved in the study.

**Data Availability Statement:** The original contributions presented in the study are included in the article, further inquiries can be directed to the corresponding author.

**Conflicts of Interest:** The authors declare that they have no competing interests.

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
