# Peer review of "Determination of Critical Factors for Success in Business Incubators and Startups in East Java"

_sustainability, doi:10.3390/su142114243_

Round 1
Reviewer 1 Report
Here we have a work posing a very interesting research question about the perception of critical success factors in the double perspective of start-ups and business incubators.
It is based on an undoubtedly time-consuming research activity, necessary to collect data to develop the empirical part of the work.
Nonetheless, from the reading it seems that it is not based on an extensive activity of international literature review. Moreover, the work misses in legibility, which consistently affects the final quality of the manuscript.
Below I identify the main weaknesses of the work and offer my comments and suggestions.
1. The manuscript is generally not well written. Some words are missing (sometimes the missing word is the subject itself), some sentences are incomplete, and one is repeated twice on pages 1 and 2 (i.e., “An agile culture combining clan with adhocracy; the ability to nurture their absorptive, innovative, and adaptive capabilities effectively; and human capital with adequate entrepreneurial skills, emotional attachment to, and fitness with the startup. It harmonizes with their agile culture, effectively enabling innovation programs enabled startups to remain focused on their innovation initiatives and not worry about scalability since the solutions collaborations between employees internally and with external actors enable rapidness to market”). Due to the writing limitations and imprecisions, some parts of the work are not clear enough. Thus, in some cases I had to read the same sentence more than once, in order to understand the point that authors were trying to make. The use of a copy editor might contribute to benefit the legibility of the paper.
2. Conducted literature review is too brief and it is not sufficiently focused on the investigated issue. I suggest authors to go much more in depth into extant literature. There is an ample international literature on the critical success factors of start-ups, which authors need to deal with in order to build the theoretical rationale of the work and to support their research choices. A specific part of the work should be expressly dedicated to it. For the work to produce a theoretical contribution, literature should clearly inform the choice of the dimensions to be investigated in the subsequent empirical part of the work and also make clear why authors do not investigate some of the critical success factors previously identified by extant literature. The issue of “digital” should be clearly illustrated in the context of the literature review as well.
3. In the “Methods” section, it is not clear how authors matched data generated through different collection procedures.
4. On page 4, in the “Results” section, authors list some elements relevant for business success, such as - for example - partnerships and coaching, but then they do not explode these things in the context of investigated critical success factors. It is not clear the reason why.
5. In Section 3.2 it should be made clear why there exist differences between the priority scales illustrated in figures 3 and 4.
6. Stemming from my previous considerations, the introduction needs substantial revision as well. I suggest you to rebuild it.
7. On page 8, in the Discussion, consider that 3 of the 4 most relevant critical success factors are common to incubators and start-ups. Moreover, as to identified factors, I think it could be useful to interpret each one of them as a whole based on its characterizing elements, as they define a same conceptual dimension.
8. In the “Conclusion” section it should be clearly highlighted why and how the work contributes to literature and authors should also identify the managerial and policy implications stemming from the work findings.
9. As a minor issue, it could be useful to reduce and better finalize the abstract. I suggest to reduce its final part, to better focus on work contribution. What is needed is a brief discussion of your findings.
10. Finally, I think the title of the work is quite different from its content and could be misleading, as you investigate the critical business success factors ACCORDING TO start-ups and incubators and not OF start-ups and incubators.
In conclusion, though interesting and ambitious in its goals and I do recognize that conducting the empirical activity has not been effortless, the contribution of the paper at present is modest and needs a consistent improvement.
The work tackles an important topic. However, in my estimation, the manuscript appears to be in its embryonic phase and still needs work and refinement.
I hope that you found my comments helpful.
Good luck as you move forward with this project.
Author Response
Thanks for your valuable comment on our manuscript
- The manuscript is generally not well written. Some words are missing (sometimes the missing word is the subject itself), some sentences are incomplete, and one is repeated twice on pages 1 and 2 (i.e., “An agile culture combining clan with adhocracy; the ability to nurture their absorptive, innovative, and adaptive capabilities effectively; and human capital with adequate entrepreneurial skills, emotional attachment to, and fitness with the startup. It harmonizes with their agile culture, effectively enabling innovation programs enabled startups to remain focused on their innovation initiatives and not worry about scalability since the solutions collaborations between employees internally and with external actors enable rapidness to market”). Due to the writing limitations and imprecisions, some parts of the work are not clear enough. Thus, in some cases I had to read the same sentence more than once, in order to understand the point that authors were trying to make. The use of a copy editor might contribute to benefit the legibility of the paper.
We Already revise this paragraph and make this clear with adjustment and more easy to read
- Conducted literature review is too brief and it is not sufficiently focused on the investigated issue. I suggest authors to go much more in depth into extant literature. There is an ample international literature on the critical success factors of start-ups, which authors need to deal with in order to build the theoretical rationale of the work and to support their research choices. A specific part of the work should be expressly dedicated to it. For the work to produce a theoretical contribution, literature should clearly inform the choice of the dimensions to be investigated in the subsequent empirical part of the work and also make clear why authors do not investigate some of the critical success factors previously identified by extant literature. The issue of “digital” should be clearly illustrated in the context of the literature review as well.
We add literature review section and use technology capability to explain the digital context in business incubator and stratup
- In the “Methods” section, it is not clear how authors matched data generated through different collection procedures.
we add some explain to relate matched data with different collection
- On page 4, in the “Results” section, authors list some elements relevant for business success, such as - for example - partnerships and coaching, but then they do not explode these things in the context of investigated critical success factors. It is not clear the reason why.
we add adjustment for this and why not include because partnership and coaching not element in government program
- In Section 3.2 it should be made clear why there exist differences between the priority scales illustrated in figures 3 and 4.
its different because figure 3 descriptive and figure 4 not
- Stemming from my previous considerations, the introduction needs substantial revision as well. I suggest you to rebuild it.
yes we already revised and add some explain about context
- On page 8, in the Discussion, consider that 3 of the 4 most relevant critical success factors are common to incubators and start-ups. Moreover, as to identified factors, I think it could be useful to interpret each one of them as a whole based on its characterizing elements, as they define a same conceptual dimension.
we add some explanation to make adjustment and discuss from the result
- In the “Conclusion” section it should be clearly highlighted why and how the work contributes to literature and authors should also identify the managerial and policy implications stemming from the work findings.
we add some managerial and policy implication
- As a minor issue, it could be useful to reduce and better finalize the abstract. I suggest to reduce its final part, to better focus on work contribution. What is needed is a brief discussion of your findings.
we reduce the abstract
- Finally, I think the title of the work is quite different from its content and could be misleading, as you investigate the critical business success factors ACCORDING TO start-ups and incubators and not OF start-ups and incubators.
yes we already adjustment for this
Reviewer 2 Report
The article presented is current, it addresses a topic much talked about and debated without great originality. The text has several serious deficiencies that make it difficult to understand what is intended.
The text does not seem to be defined as objectives, if we read the abstract it is not clear what gap or problem the authors intend to answer, since the main focus is on the methods, which are mere tools to reach the intended purpose, which we do not know what it is. The abstract needs to be rewritten
As there is no literature review chapter, the entire article is completely without scientific support. We do not know which theory or theories underlie the development of the work, we do not know that other similar works were carried out. Also, we do not know if the present work complements or tries to give a new approach and what the most recent works in this area are.
The only focus of the article seems to be the tool, the quantitative approach, which is of little use when we don't know what the starting point is.
There is also no discussion, because there is no theoretical basis for comparing the data obtained. As for conclusions, nothing is possible to conclude.
The text to be accepted needs a serious and profound revision.
Author Response
Thanks for your valuable comment,
we revised abstract to make it clear, we explain the context in introduction with research gap. as you suggest we add literature review, adjustment method, and add some explanation in discussion based on result
Reviewer 3 Report
The paper presents a good case report but, in my opinion, will be necessary to create a new section that presents the state-of-the-art in a systematic way. This work is very relevant, but it is necessary to know your contribution/innovation aspects considering the other works in the area. Maybe the comparison with works that describe similar geographical regions and industries/companies.
Author Response

(The authors gave the same response as above.)

Round 2
Reviewer 2 Report
The manuscript has been substantially improved and can be published
Author Response
Respected Reviewer,
The paper has been revised and modified as per your suggestions. Thank you very much for your review and valuable suggestions.
Thank you